# Repurposing Vitamin C for Cancer Treatment: Focus on Targeting the Tumor Microenvironment

**DOI:** 10.3390/cancers14112608

**Published:** 2022-05-25

**Authors:** Wen-Ning Li, Shi-Jiao Zhang, Jia-Qing Feng, Wei-Lin Jin

**Affiliations:** 1The First Clinical Medical College of Lanzhou University, Lanzhou 730000, China; liwn18@lzu.edu.cn (W.-N.L.); zhangshj2018@lzu.edu.cn (S.-J.Z.); fengjq18@lzu.edu.cn (J.-Q.F.); 2Institute of Cancer Neuroscience, Medical Frontier Innovation Research Center, The First Hospital of Lanzhou University, Lanzhou 730000, China

**Keywords:** vitamin C, anti-immunity, dietary intervention, drug repurposing, tumor microenvironment

## Abstract

**Simple Summary:**

The tumor microenvironment (TME) is a complicated network, and several promising TME-targeted therapies, such as immunotherapy and targeted therapies, are now facing problems over low response rates and drug resistance. Vitamin C (VitC) has been extensively studied as a dietary nutrient and multi-targeted natural drug for fighting against tumor cells. The focus has been recently on its crucial functions in the TME. Here, we discuss the potential mechanisms of VitC in several specialized microenvironments, characterize the current status of its preclinical and clinical applications, and offer suggestions for future studies. This article is intended to provide basic researchers and clinicians with a detailed picture of VitC targeting the tumor microenvironment.

**Abstract:**

Based on the enhanced knowledge on the tumor microenvironment (TME), a more comprehensive treatment landscape for targeting the TME has emerged. This microenvironment provides multiple therapeutic targets due to its diverse characteristics, leading to numerous TME-targeted strategies. With multifaced activities targeting tumors and the TME, vitamin C is renown as a promising candidate for combination therapy. In this review, we present new advances in how vitamin C reshapes the TME in the immune, hypoxic, metabolic, acidic, neurological, mechanical, and microbial dimensions. These findings will open new possibilities for multiple therapeutic avenues in the fight against cancer. We also review the available preclinical and clinical evidence of vitamin C combined with established therapies, highlighting vitamin C as an adjuvant that can be exploited for novel therapeutics. Finally, we discuss unresolved questions and directions that merit further investigation.

## 1. Introduction

Cancer is a complex systemic disease, and current strategies for cancer treatment combine surgery, radiotherapy, chemotherapy, and other modalities. The tumor microenvironment (TME) is a highly heterogeneous micro-ecosystem composed of surrounding immune cells, fibroblasts, vessels, and the extracellular matrix (ECM). It provides constant growth-stimulating signals and nutritional support to the cancer cells embedded [1]. Therefore, the TME is considered as a new therapeutic target to inhibit tumor growth, metastasis, and drug resistance. At present, the pattern for cancer treatment has gradually switched from a cancer-centric model to a TME-centric one. Of these, immunotherapy and targeted therapies have achieved impressive success, but are also obstructed by various barriers. Immunosuppression is one of the most important hallmarks of TME. Tregs, tumor-associated macrophages (TAMs), and myeloid-derived suppressor cells (MDSCs) infiltrate into the TME and secrete large amounts of immunosuppressive mediators. This process contributes to the tumor effector cell depletion and M1-to-M2 phenotype conversion. In addition, poor angiogenesis is often related to intratumoral hypoxia, low pH, and high pressure. Based on the characteristics of the TME, our laboratory has recently divided the whole complex TME into six specialized microenvironments, namely, immune microenvironment, metabolic microenvironment, hypoxic niche, acidic niche, innervated niche, and mechanical microenvironment [2]. Classifying the TME according to its characteristics paves the way for further designs of targeted therapeutic strategies. Growing evidence suggests that the gut and tumor microbiota and their metabolites play complex roles in tumorigenesis and treatment responsiveness, both directly and indirectly. Thus, the microbial microenvironment is emerging as a seventh specialized TME that determines the direction of cancer progression [3]. These specialized microenvironments crosstalk with each other and interact with the whole organism to form a cancer ecosystem (Figure 1).

Vitamin C (VitC) (or ascorbic acid (AA)) is not only an essential dietary nutrient, but a natural agent with multiple therapeutic properties. However, the debate over whether VitC can combat cancer has been going on for decades, although results remain uncertain and controversial [4,5]. In recent years, preclinical studies have uncovered the novel mechanisms of VitC in epigenetic, immunomodulatory, and selective cytotoxicity. Hence, VitC is back to the forefront as an adjuvant to assist various cancer treatments. Repurposing VitC is an effective and promising combination option to target the TME.

In this review, we briefly describe the anticancer potential of VitC as a repurposed drug and then we explain the novel mechanisms of how VitC targets the specialized microenvironments. Finally, we summarize and discuss the most recent advances in its monotherapy and adjuvant therapy.

## 2. VitC Is an Example of Repurposed Drugs with Anticancer Activity

Drug repurposing has become an attractive concept due to its low cost and high safety profile. VitC shows outstanding anticancer potential among related conventional drugs. Chen et al. recently implemented an umbrella review to assess the association between VitC intake and cancer incidence and prognosis. This study of 3562 articles (from systematic reviews and meta-analyses) on 22 tumors has demonstrated that VitC intake was negatively associated with the incidence of 11 tumors in multiple organ systems, along with better cancer outcomes in breast cancer patients when supplemented with VitC [6]. Consistent with this conclusion, another umbrella study that included 76 meta-analyses has shown that VitC supplementation resulted in dose-dependent reductions in all-cause mortality and the risk of various cancers [7]. However, two Nurses’ Health Studies conducted over 32 and 22 years, respectively, showed no association between VitC intake and risk of breast cancer. This finding is consistent with most previous prospective studies [8]. The current epidemiological evidence for VitC preventing or treating tumors is inconsistent, requiring additional support from laboratory and clinical studies.

A large and gradually increasing number of cell lines, animal, and clinical studies have reported positive results (Table 1). In various cancer cell lines, especially lymphoma, the pharmacological dose of VitC acts as an H_2_O_2_ pro-drug in tissues, selectively inducing apoptosis or necrosis in cancer cells without impairing normal cells [9]. In KRAS and BRAF mutant colorectal cancer cells, Yun et al. elucidated that dehydroascorbic acid (DHA) enters the cell via GLUT1 and is then reduced to AA at the expense of GSH oxidation and GAPDH inactivation. This phenomenon leads to lethal oxidative damage and energy crisis [10]. For the hematologic tumors and some solid tumors, such as melanoma, VitC inhibits tumor growth and even reverses malignant transformation through epigenetic regulation of gene expression profiles [11,12]. Melanoma and breast cancer in VitC-deficient mice exhibited a higher propensity for invasiveness and metastasis, while restored VitC inhibited those malignant behaviors [13]. In ovarian cancer, the combination of VitC with carboplatin and paclitaxel suppresses xenograft growth and possesses considerably stronger efficiency than chemotherapy alone. Intravenous VitC (IVC) also alleviates chemotherapy-related adverse effects in clinical patients [14]. In addition, many completed and ongoing randomized controlled trials have been designed to explore the impact of IVC or oral VitC on cancer treatment and prognosis (Available online: https://clinicaltrials.gov (accessed on 20 May 2022)).

These results suggest that the biological effects of VitC are apparently dose-dependent, with different concentrations achieved by oral or parenteral administration, exerting multifaceted anticancer effects. In terms of transport mechanisms, the transmembrane transporters SVCTs and GLUTs transport ascorbic acid and DHA, respectively, and promote intracellular accumulation in tumor cells and other normal cells. The expression levels of these two transporters vary by organ and correspond to plasma concentrations, metabolic needs, and some specific gene mutations [33]. The mechanism underlies the functions as a cofactor to potentiate HIF, TET, JHMDs, and other hydroxylases, thereby involving a variety of enzymatic reactions (Figure 2).

## 3. VitC Targets Not Only Cancers but also the TME to Exert Anticancer Activity

Most previous studies have focused on the role of VitC in targeting tumor cells in isolation. As the effects on neighboring non-tumor cells and their secretions emerge, studies on the mechanisms underlying the specialized tumor microenvironment will contribute to a deeper understanding of the impact of VitC in the complex microenvironment (Figure 3).

### 3.1. VitC and Immune Microenvironment

VitC regulates epigenetic processes and signaling pathways to further reprogram the tumor immune microenvironment (TIME). Numerous studies have investigated the potential of VitC in modulating immunoreactive and immunosuppressive cells and their secretion. Hence, VitC is considered as a promising drug to turn cold cancer (immunosuppressive) into hot ones (immunoreactive).

VitC may reverse the immunosuppressive TME at the cellular and molecular levels. VitC increases the infiltration of T cells in TME and promotes immune memory formation. Further studies have found that the infiltrated T lymphocytes have an elevated CD8^+^/CD4^+^ ratio, reflecting a significant improvement in the antitumor effect [18,34]. During early NK cell differentiation, VitC increases the frequency of KIR gene expression, a mature NK cell marker, by enhancing the demethylation of the KIR promoter [24]. High-dose VitC is also observed to enhance the secretion of perforin and granzyme B by activated NK cells [35]. TAMs are converted to an M2 phenotype in response to immunosuppressive cytokines within the TME, thereby promoting cancer to overcome immune surveillance. Xu et al. demonstrated that VitC induces apoptosis of infiltrating M2-type TAM in tumor nodules and inhibits the epithelial–mesenchymal transition (EMT) in a murine epithelial ovarian cancer model [26]. Recent findings by Ang et al. suggest that monocyte phenotype-related gene and protein expression is influenced by intracellular VitC availability. Although the exact effects are yet to be determined, the authors find some alterations in the M2-associated gene expression and protein secretion when VitC is added [25]. Tregs cells are regarded as the primary cells that maintain a state of immune tolerance. Someya et al. reported that VitC decreases the methylation of CNS2 to promote Foxp3+ expression, which implies a robust immunosuppressive capacity [36]. Interestingly, the derivative pVitC induces the expression of the Tregs cell marker Foxp3+ in Vγ9Vδ2 T cells, which is also epigenetically regulated by TET and in the presence of TGF-β [20]. These results show the complexity of VitC targeting TIME. VitC also affects the function of other innate immune cells (e.g., DCs and neutrophils), such as attenuating neutrophil extracellular trap formation (Table 1). Moreover, immunosuppressive cell products (e.g., IL6, IL17, and lactate) in the microenvironment are also subjected to VitC, and these products play an essential role in the TIME remodeling [13,22,37].

### 3.2. VitC and Metabolic Microenvironment

Under the nutrient-limited microenvironment, cancer cells and stromal cells have a fierce competition for limited energy sources of glucose and glutamine. Aerobic glycolysis (Warburg effect) and glutamine addiction confer a selective growth advantage to cancer cells, whereby inhibiting surrounding stromal cell activity. VitC antagonizes the HIF-driven glucose metabolic reprogramming from OXPHOS to glycolysis in various cancer cells. Meanwhile, VitC promotes glutamine synthetase degradation by inducing redox imbalance, thus attenuating endogenous glutamate-dependent tumor growth in vitro and in vivo [38,39]. Therefore, VitC may influence the nutrient partitioning in TME, whereas the effect on other TME cells in a tumor context needs to be further explored.

Beyond energy metabolism, the precise regulation of intracellular and extracellular ROS levels also decides the fate of tumor cells. Although the oxidative defense system within tumor cells can precisely regulate ROS levels within the appropriate range, high-dose VitC uses multiple mechanisms to disrupt the system and destroy the tumor. Accumulation of ROS in fibroblasts contributes to the conversion to a cancer-associated fibroblast phenotype, which pushes the tumor toward invasiveness and progression [40]. The physiological dose of VitC prevents radiation-induced myofibroblast phenotype by scavenging the ROS in the fibroblast [41]. The ECM is also under constant oxidative stress due to the chronic inflammatory stimulation in cancer. Cells, such as Tregs and MDSCs, can release ROS into the microenvironment, while elevated ROS promotes tumor growth and angiogenesis and stimulates immunosuppressive cell proliferation in turn [42]. ROS also plays a critical role in reshaping the ECM and modulating the ECM-tumor cell interface, especially during the ECM detachment [43]. Using the antioxidant properties of VitC to fight tumors is controversial because studies have found that some tumors can upregulate antioxidants to neutralize the ROS and alleviate oxidative stress.

### 3.3. VitC and Hypoxic Microenvironment

Hypoxia is commonly observed in solid tumors due to a mismatch between vascularization and tumor growth. Tumor angiogenesis tends to be leaky and discontinuous under the hypoxic microenvironment. This phenomenon promotes distant tumor metastasis while hindering immune cell infiltration and drug delivery [44]. VitC has been shown to both inhibit abnormal angiogenesis and promote vessel normalization. Orally administered high-dose VitC suppressed tumor angiogenesis in colon cancer-bearing mice by downregulating the expression and secretion of VEGF-A and VEGF-D [45]. VitC impairs COX-2 activity and inhibits VEGF mRNA expression in melanoma cells in a time-dependent manner [46]. In terms of vascular architecture, VitC attenuates vascular permeability by inducing endothelial cell retraction and epigenetically preventing apoptosis of endothelial cells [37,47].

The hypoxic microenvironment also provides unique opportunities for IVC therapy. Hypoxia-stimulated high HIF expression increases the sensitivity of cancer cells to VitC [48]. For example, abnormal tumor vascularization prolongs the residence time of VitC inside the tumor. GLUT1 induced by HIF increases intracellular DHA accumulation and renders cancer cells more sensitive to VitC treatment. HIF-independent hypoxic adaptation of cancer cells also increases cancer vulnerability. The unfolded protein response (UPR) is activated to relieve the endoplasmic reticulum stress (ER) in response to severe and persistent hypoxia. If UPR is insufficient to alleviate excessive ER stress effectively, it will induce cancer cells to undergo apoptosis. VitC increases the ER stress-mediated breast cancer apoptosis via activation of the IRE-JNK-CHOP signaling pathway, an effect independent of ROS [49]. Accordingly, the overexpression of HIF, GLUTs, and the elevated ER stress may serve as biomarkers to predict the efficacy of VitC treatment.

### 3.4. VitC and Acidic Microenvironment

Depending on the intracellular buffering system and exocytosis system, tumors can maintain stable intracellular pH (pHi ≥ 7.4) and lower extracellular pH (pHe = 6.7–7.1) under metabolic stress conditions. This phenomenon is known as the pH inversion [50]. Dysregulated pH affects almost every aspect of malignancy, including immunosuppression, stromal degradation, metastatic spread, and drug resistance. 

VitC acts as a glycolysis inhibitor to reduce endogenous acidic metabolic products, mainly lactate and H+. On the one hand, VitC directly inhibits HIF-1α-mediated glycolysis-related genes expression and the downstream acidic metabolites [51]. On the other hand, the ROS generated by VitC treatment exerts a synergistic effect with other glycolysis inhibitors, providing a combined therapeutic strategy [15,52]. Hypoxia-induced carbonic anhydrase 9 (CAIX) and monocarboxylate transporter (MCT) are also two crucial downstream factors of HIF, which catalyze CO_2_ hydration to produce protons and drive H+ and lactate export. A retrospective analysis of patient-derived breast cancer tissues showed that the low-dose VitC group expressed higher levels of HIF and CA-IX than the high-dose group, implying a higher malignancy grade and poor prognosis [53]. The proton pump inhibitor (PPI) as a repurposing drug can inhibit V-ATPase, which in turn regulates the intracellular and extracellular pH gradient and resists acidic microenvironment-induced drug resistance. PPI has also been reported to reduce the release of exosomes from melanoma cells in response to low pH stimulation [54]. Li et al. demonstrated that VitC synergizes with pantoprazole to overcome drug resistance in prostate cancer and inhibit exosome secretion [55].

The acidic microenvironment has recently been found to act in turn on VitC homeostasis. Some factors, such as pH gradient and local potential, may affect the passive diffusive transport of VitC and regulate intracellular concentrations [56]. This pattern of transcellular transport, although not the dominant one, provides a plausible explanation for the cancer-selective toxicity of VitC.

### 3.5. VitC and Innerved Niche

Evidence to support the effect of VitC on the innerved niche is currently lacking, but one possible mechanism is as a neuromodulator. The protumor effect of norepinephrine is one of the relatively explicit findings in this under-studied field [57]. Biosynthesis of the nerve cell norepinephrine is dependent on dopamine β hydroxylase, and VitC can act as a cofactor for the enzyme to improve activity [58]. Furthermore, VitC modulates synaptic plasticity and facilitates the storage and release of other neurotransmitters, such as dopamine, acetylcholine, glutamate, neuropeptides [59,60,61].

In addition to neurons, glial cells, such as Schwann cells and oligodendrocytes, play a non-negligible role in neuro-tumor communications. A recent example in pancreatic cancer indicates that the autophagy of Schwann cells is activated and promotes perineural invasion (PNI) [62]. Collagen synthesis induced by proline hydroxylase was previously inferred to be the primary mechanism of VitC promotion of myelin formation. Until recently, Huff et al. reported that VitC increases DNA demethylation of genes involved in the myelination of Schwann cell and the formation of ECM [63].

VitC is highly enriched in the neurons, suggesting its crucial role in neurobiology. VitC can promote neural development, promote neurite growth, and mediate the metabolic coupling between glial cells and neurons through the AA cycle [64,65]. Contrary to previous results, Ferrada et al. demonstrated in an in vitro study that physiological doses of VitC (200 μM) induced necrotizing apoptosis in neurons [32]. Collectively, these existing studies at least indicate that VitC is necessary for the neurobiological function in the TME.

### 3.6. VitC and Mechanical Microenvironment

Cancer metastasis is a complex and multistep process involving a series of malignant events. During the early stages of metastasis, cancer cells can sense changes in the stiffness of ECM and transmit mechanical signals into the cell, triggering cytoskeletal rearrangement. VitC plays an important role in the mechanical microenvironment, regulating from stromal cells and ECM to cellular mechanical signaling.

The stiffness of the ECM depends mainly on the balance between collagen production and protease-mediated degradation, which affects the matrix integrity and coordinates cancer metastasis [66]. VitC enhances the hydroxylation at the proline and lysine residues as a cofactor of collagen prolyl and lysyl hydroxylase to form the correct triple-helical conformation [30]. VitC inhibits a variety of metalloproteinases (MMPs) mRNA, which degrade ECM and release growth factors that drive tumor metastasis [13,67]. Cancer-associated fibroblasts (CAFs), the primary stromal cells to constitute the ECM, are prominent in the ECM remodeling by secreting multiple types of collagen and MMPs. Compared with the normal fibroblasts, CAFs often obtain immortalized features and enhanced secretory capacity to promote tumor invasion. VitC treatment altered the gene expression profile in immortalized mouse embryonic fibroblasts, where genes associated with collagen synthesis, cell adhesion, and extracellular matrix were significantly downregulated. More importantly, VitC exhibited a dose-dependent dual effect on immortalized MEF, similar to cancer cells [68]. VitC also inhibits EMT by controlling integrin and YAP/TAZ mechanistic signaling. Collagen proline hydroxylation has been shown to enhance affinity with integrins directly or indirectly [69]. VitC derivatives increase the phosphorylation of ERK1/2, which may be involved in integrin-mediated signaling pathways during cell stretch [70]. SYNPO-2 has been reported to inhibit the activity of YAP/TAZ in the Hippo signaling pathway [71]. Gan et al. demonstrated that VitC treatment reduces YAP1 expression while upregulating SYNPO-2; therefore, inhibiting metastasis of TNBC via actin dynamics modulation [72]. Overall, the biomechanics and mechanical signals of ECM may be a target for VitC to hamper malignant invasion and metastasis.

### 3.7. VitC and Microbial Microenvironment

The microbial microenvironment is an ecological network consisting of the intestinal microbiota and the intratumor microbiota, as well as their metabolites. In particular, the immune-oncology-microbiota axis (IOM) is one of the most promising directions and a therapeutic target.

VitC regulates the intestinal microbial homeostasis. Pham and Otten et al. independently conducted two pilot studies exploring the effects of vitamin-targeted administration on the flora composition and metabolism. Pham et al. utilized macrogenome sequencing and found that the diversity of microbiota α and the relative abundance of fecal *Collinsella* were increased compared with pretreatment, accompanied by increased short-chain fatty acids (SCFAs), butyrate metabolites, and decreased pH [73]. Unlike the above, Otten et al. did not directly assay gut metabolites, but they found a significant increase (*p* < 0.05) in the relative abundance of *Lachnospiraceae* in the flora, a microbe that also produces SCFAs [74]. The proportion of strictly anaerobic bacteria has been linked to the intestinal redox potential, and this finding may help to explain the microbiota alterations. Given that bacterial metabolites, such as SCFAs, can enhance the immunogenicity of situ and a variety of distant cancers, the results open new possibilities for VitC to enhance anticancer immunity [75].

VitC is also a parameter that influences microbes within the tumor. First, VitC plays a crucial role in preserving the integrity of the physical barrier against microbial spreading and colonization [31]. Moreover, increased retroviral transcription in tumor cells by VitC appears to amplify the efficacy of anti-PD-1 immune checkpoint therapy (ICT) and epigenetic therapy [35,76]. Oncolytic virus-induced immunogenic cell death (ICD) is anticancer immunotherapy that stimulates tumor-specific immune responses by releasing immunoreactive substances from dying tumor cells. Ma et al. reported that ROS derived from high doses of VitC enhances OVs-mediated ICD and somewhat reverses the suppressive immune microenvironment [77].

These direct and indirect results also yield an appealing scenario: dietary or VitC supplements may target intestinal microecology to synergize with antitumor therapy.

### 3.8. The Interplay between VitC and the Complicated Metastatic TME

To be precise, the TMEs are more like a complex whole with distinct organ features rather than separated compartments. Metastatic tumor foci retain the characteristics of the primary tumor, while also evolving to accommodate inefficient metastatic processes and new target organs. For instance, bone is one of the most common sites of metastasis for multiple advanced tumors, exhibiting complex and unique features in immunosuppression, hypoxia, and ECM remodeling. There has been considerable evidence to support the role of VitC in bone metabolism, thus it is worth exploring whether VitC affects bone metastases.

During bone metastases, cellular compounds produced by the primary tumor enter the bloodstream and act at distant sites, stimulating angiogenesis and infiltration of Tregs, MDSCs, and other immune cells to generate a pre-metastatic niche. The colonized tumor cells then continuously interact with microenvironments to promote skeletal remodeling and immunosuppression, culminating in overt bone metastases [78]. In addition, hypoxia and ECM remodeling are engaged in the process. By epigenetic mechanisms, VitC significantly improves bone metabolism and bone formation. VitC modulates stromal cell differentiation, promoting osteoblast development while suppressing osteoclasts, possibly alleviating tumor-induced aggressive osteolysis [79,80]. The synthesis of type I collagen is enhanced by VitC, which not only improves the stability of the bone matrix but also contributes to osteoblasts differentiation [81]. Furthermore, Kolke et al. demonstrated that higher doses of VitC (100 μM) altered miRNA expression to up- or down-regulate signaling pathways associated with cell adhesion, differentiation, and cell stemness in BMSCs [82]. RANKL/NFκB in osteoclasts is an essential signaling pathway that accelerates bone metastasis. Hie et al. reported that VitC deficiency status upregulates RANK expression in vitro and in vivo, suggesting a negative regulation of RANK [83]. These findings indicate that VitC may inhibit tumor growth by acting on metastases, as well as the main location.

The existing research on the effects of VitC on the bone microenvironment focuses mostly on preventing osteolysis and bone loss. Further research into this function for certain osteogenic metastases, such as prostate cancer, is warranted [84]. Anyway, VitC may help patients with advanced cancers avoid complications like osteoporosis and fractures by acting as a supportive treatment.

## 4. Application of VitC as a Single Agent or Adjuvant to Target the TME

As the routes of administration correspond to different bioavailabilities and by explaining the controversy over the studies of Cameron and Pauling, interest in the anticancer ability of VitC has been revived [85]. We will present the latest applications of VitC in single-agent and combination strategies against tumors (Table 2).

### 4.1. Monotherapy

Although VitC alone has been reported to inhibit the viability of various solid and hematologic cancers in preclinical studies, several phase I/II clinical studies did not obtain supportive results from IVC therapy [100,101,102]. Nevertheless, monotherapy is an excellent option to investigate the safety and adverse effects of VitC therapy, as it can exclude drug–drug interferences and assess the therapeutic effect. A high-dose of IVC is well-tolerated by most patients, with only mild adverse effects. The most frequent adverse reactions include nausea, diarrhea, dryness, and headache, and these effects are usually resolved at the end of treatment [101,103]. However, severe adverse reactions, such as hemolysis and kidney stones in a minority of patients, should be vigilantly monitored. VitC monotherapy in palliative care alleviates cancer-related pain and fatigue and improves the patients’ quality of life [104]. The results in monotherapy underlie the rationale and feasibility of combination therapy. Large-scale randomized clinical trials need to be performed in the future to rule out placebo effects, predict responsiveness across patients, and identify the optimal dose to be administered.

### 4.2. Combination Therapy

VitC is widely being explored as an adjuvant treatment strategy. Its combination with conventional radiotherapy and chemotherapy amplifies the cytotoxicity to cancer cells and reduces the treatment-related side effects and off-target effects. Several reviews have summarized the research in this area [105,106]. Herein, we will focus on combining VitC with novel therapies in cancers and TME.

For targeted therapy, the multifaceted nature of VitC facilitates its combination with different strategies. Future therapeutics could benefit from a combination of VitC and anti-angiogenic drugs, such as small molecule multi-kinase inhibitors and VEGF monoclonal antibodies. VitC enhances the killing efficiency of Hep G2 cells by low-dose sorafenib in vitro. The authors also reported a patient with rib metastases who achieved more prolonged remission after receiving combination therapy [90]. To assess the efficacy and safety of IVC in combination with erlotinib and gemcitabine, a Phase I clinical trial was implemented in patients with metastatic pancreatic cancer. The addition of VitC did not increase adverse effects but alleviated tumor progression [91]. Bevacizumab inhibits tumor growth by blocking the binding of VEGF to the vascular endothelial receptor, which targets the tumor vascular system. Three clinical trials are performed to investigate the impact of VitC combined with bevacizumab and chemotherapy drugs to treat advanced cancer patients (NCT01891747; NCT02969681; NCT04516681). Cancer stem cells (CSCs) present a subpopulation with stem-cell-like properties of self-renewal and differentiation, which are considered as the root of cancer recurrence and drug resistance. High-dose VitC may selectively kill CSCs through oxidative damage and modification of pluripotency genes [107,108]. Several recent studies have suggested that VitC acts synergistically with antibiotics (e.g., doxycycline and azithromycin) by targeting the mitochondria of CSCs to inhibit cell proliferation and invasion [93,94]. These studies provide a synthetic-lethal metabolic strategy to target specific cancer cells. Like VitC, metformin is a versatile anticancer agent that targets energy metabolism, stimulates immunity, and reshapes the TME. A combined clinical trial is currently underway to evaluate the synergistic effects of VitC and metformin (NCT04033107). Besides, several studies have also focused on oral VitC enhancing the anti-tumor efficacy of epigenetic agents. In mice bearing triple-negative breast cancer and melanoma, VitC supplementation improves sensitivity to BETi therapy [96,97]. In a randomized, double-blind trial in patients with bone marrow cancer, oral VitC promoted DNA demethylation in a synergistic manner with DNMTi therapy (5-azacytidine) [76].

For immunotherapy, the combination strategy yielded some positive results. Luchtel et al. first combined high-dose VitC with anti-PD-1 therapy and found increased infiltration of CD8^+^ T, DC, and NK cells alongside enhanced cellular activity in murine B-cell lymphomas [35]. In general, combination therapy yields considerably better tumor suppression than monotherapy. Magrì’s et al. subsequently obtained encouraging results by using high doses of VitC in combination with anti-PD-1 or anti-CTLA-4 for colon, breast, melanoma, and pancreatic cancers. VitC boosts the efficacy of ICT against drug-resistant cancers, even eliminating some breast and colon cancer cells and preventing cancer recurrence [18]. Mechanistically, VitC does not directly alter the PD-L1 mRNA but increases the IFN-γ-induced chemokines in a TET-dependent manner, ultimately acting on the PD-L1 gene [34]. Supporting this conclusion, Peng et al. recently revealed that VitC stimulation of TET2 activity in the renal cell carcinoma significantly increases T-lymphocyte infiltration and indirectly affects PD-L1 expression via the IFN-γ/STAT1/IRF1 signaling pathway [86]. Furthermore, the role of VitC as a promoter of γδ T-cell, NK cell immunotherapy is also under investigation [21,109]. Future studies will focus more on VitC as an adjuvant to ICT and adoptive cell therapy.

Regarding nutrition and metabolism, several studies have attempted to explore IVC with nutritional therapy in cancer patients. Fasting-mimicking diet and ketogenic diet postpone tumor progression by depleting glucose in TME and improving immunity. Di Tano et al. demonstrated that fasting-mimicking diet and VitC treatment exerted a synergistic effect on oxidative damage and chemotherapy-induced cytotoxicity in KRAS-mutant tumors [99]. In another study, patients who received IVC after ketogenic therapy had increased levels of ketone bodies and reduced generalized inflammation compared to pretreatment [98]. Despite the encouraging results, the safety of this calorie restriction strategy remains disputable, considering cancer patients often experience malnutrition and increased energy expenditure.

## 5. Conclusions

We have provided a multilevel, multifaceted perspective on the anticancer activity of VitC. We have discussed many aspects, such as tumor immunity, metabolism, neuromodulation, and the microbiome. A large body of preclinical and clinical evidence shows great prospects for the therapeutic application of VitC, especially with immunotherapy, targeted therapies, and dietary therapies. Despite new advances in the anticancer mechanisms of VitC, some challenges that deserve in-depth investigation: (1) whether the antioxidant properties of VitC promote tumor diffusion; (2) whether VitC could be combined with dietary therapies, such as ketogenesis and fasting; and (3) the effect of VitC on the microbiome and bacterial-derived VitC on efficacy. Cancer is not static but a multistage, dynamic process, and VitC may play different roles at different stages. The value of VitC in anticancer therapy has reemerged, urging more research to exploit its potential in targeting the TME and cancer cells in the future.

## Figures and Tables

**Figure 1 cancers-14-02608-f001:**
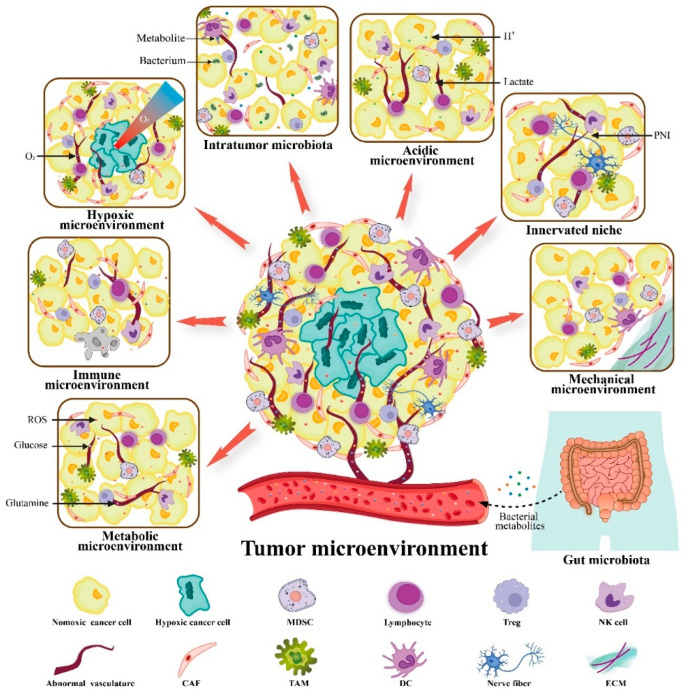
Sources of tumor microenvironment (TME). The TME has been divided into seven specialized microenvironments to investigate the tumor–stroma interactions: metabolic microenvironment, immune microenvironment, hypoxic microenvironment, acidic microenvironment, innervated niche, mechanical microenvironment, and microbial microenvironment (containing gut and intra-tumor microbiota). These specialized microenvironments engage in crosstalk to work together on the tumor and the entire organism. Adapted from reference [2]. MDSC, myeloid-derived suppressor cell; CAF, cancer-associated fibroblast; TAM, tumor-associated macrophage; DC, dendritic cell; ECM, extracellular matrix; PNI, perineural invasion.

**Figure 2 cancers-14-02608-f002:**
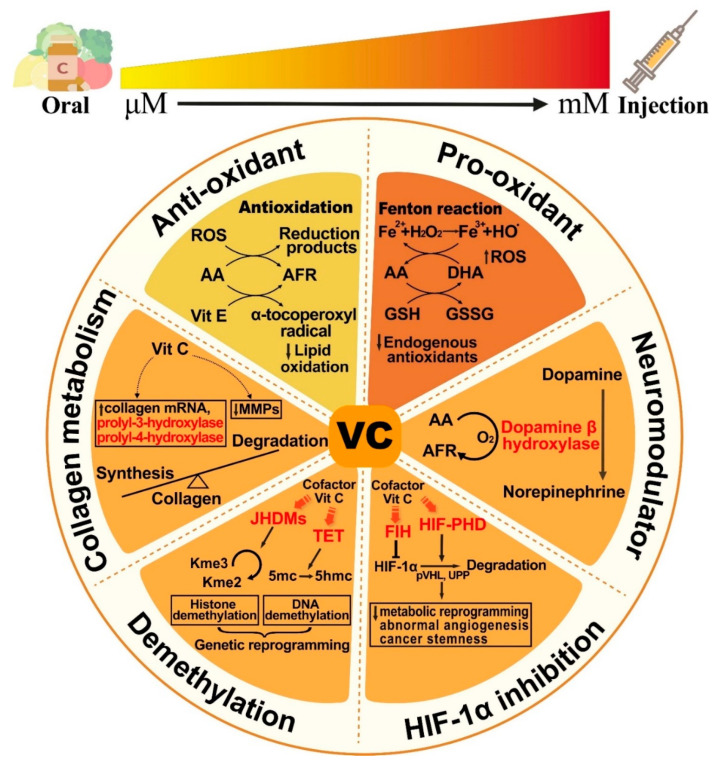
Vitamin C is a multifunctional natural nutrient that exhibits dose-dependent effects. Vitamin C (VitC) at the physiological dose (μM) is known to exhibit antioxidant properties. However, it functions as a prooxidant at the pharmacological dose (mM) achieved by intravenous administration. VitC also enhances a range of intracellular enzymatic reactions by serving as a cofactor of monooxygenases (e.g., dopamine β hydroxylase) and Fe (II)- and 2-oxoglutarate (2-OG)-dependent dioxygenases (e.g., HIF, TET, JHDMs, P-3-H, and P-4-H). Possible anticancer mechanisms include triggering oxidative damage, regulating epigenetics, blunting adaptive responses to hypoxia, and synthesizing collagen and neurotransmitters. AFR, ascorbate free radical; DHA, dehydroascorbic acid; MMPs, matrix metalloproteinases; GSH, glutathione; GSSG, glutathione disulfide; P-3-H, prolyl-4-hydroxylases; P-4-H, prolyl-4-hydroxylases; JHDMs, Jumonji-C domain-containing histone demethylases; TET, ten-eleven translocation; Kme3, trimethyl lysine; Kme2, dimethyl lysine; 5mc, 5-methylcytosine; 5hmc, 5-hydroxymethylcytosine; FIH, factor inhibiting HIF; HIF-PHD, HIF-prolyl hydroxylase; pVHL, VHL tumor suppressor protein; UPP, ubiquitin–proteasome pathway.

**Figure 3 cancers-14-02608-f003:**
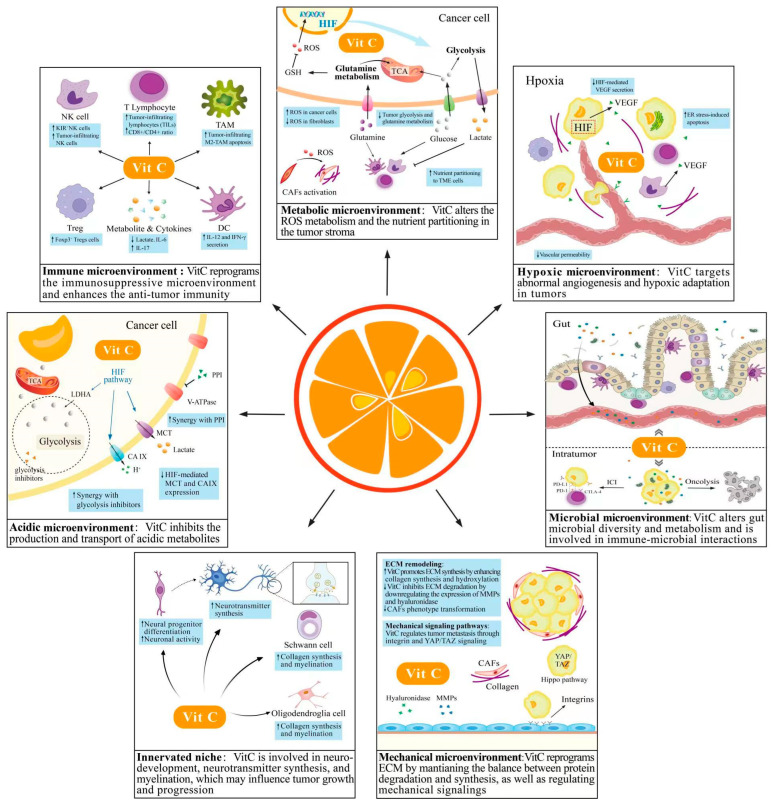
Underlying mechanisms of Vitamin C targeting the specialized tumor microenvironments. The interplay between tumor cells and the TME promotes an aggressive phenotype in various ways, including immune tolerance, metabolic reprogramming, angiogenesis, and tumor innervation. Vitamin C (VitC) with multi-targeted effects may reverse the tumor-promoting microenvironments, displaying a wide range of anticancer activities.

**Table 1 cancers-14-02608-t001:** Selected studies on the effects of different doses of VitC to target cancer cells and TME.

Category		Study Type(s)	VitC Dose and Administration	Main Mechanism	Findings	Reference
cancer cells	Multiple cancers	cell line	0–20 mM	pro-oxidant	Pharmacologic VitC selectively kills multiple cancer cells by initiating the production of extracellular H_2_O_2_	[9]
	KRAS or BRAF mutant CRC	cell line and animal	0–3 mM (in vitro); 4 g/kg, i.p. (in vivo)	pro-oxidant	DHA, the oxidized form of VitC, exhibits selective toxicity by elevating ROS to disrupt cancer cell metabolism	[10]
	Breast cancer	cell line	0–10 mM	pro-oxidant	VitC dose-dependently regulates the p66Shc/Rac1 pathway, which in turn induces apoptosis through ROS overexpression in cancer cells	[15]
	Melanoma	cell line	0–2 mM	DNA demethylation	Physiological concentrations of VitC inhibit melanoma migration and malignant transformation by increasing 5hmC levels without damaging normal melanocytes	[12]
	Melanoma, Breast cancer	animal	500 ppm pVitC, 150 mg/L VitC, oral	hydroxylase cofactor	Oral VitC promotes tumor collagen encapsulation and reduces MMP-9, IL-6, and VEGF levels, thereby inhibiting tumor growth and metastasis	[13]
	VHL-deficient ccRCC	animal	2 g/kg, i.p.	expression of HIF target genes is suppressed by enhanced TET2 activity	VitC inhibits HIF1/2α-mediated tumor metabolic reprogramming in a TET2-dependent manner, and increases the efficiency of glycolysis inhibitor (2-DG) to suppress ccRCC	[16]
	TET2 and TP53 mutant Leukemia	cell line	0–500 μM	DNA demethylation	VitC inhibits the proliferation of SKM-1 cells and promotes their differentiation to monocytes by restoring 5hmC levels	[17]
	Leukemia	cell line and animal	250 μM (in vitro); 4 g/kg, i.p. (in vivo)	DNA demethylation	VitC reverses aberrant AML self-renewal and promotes myeloid differentiation through the restoration of TET2 and TET3	[11]
tumor stromal cells	T Lymphocytes	animal	4 g/kg, i.p.	-	IVC promotes T cell differentiation, maturation, and immune memory formation, thereby increasing intra-tumor infiltration and immune responsiveness	[18]
	CD4^+^ Tregs cells	cell line and animal	100 μg/mL (in vitro)	DNA demethylation	VitC was shown to enhance the expression and stability of Foxp3+ markers in a TET2/3-dependent manner during iTregs cell differentiation	[19]
	γδ T cells	cell line	0–200 μg/mL pVitC	DNA demethylation	The derivative pVitC regulates TGF-β-induced γδ T cell expansion and promotes conversion to Foxp3^+^ Tregs cells	[20]
	γδ T cells	cell line	12.5 μg/mL VitC, 50 μg/mL pVitC	-	High concentrations of VitC and pVitC promote restimulated Vγ9Vδ2 T cell expansion through an accelerated cell cycle and affect Th1/Th2 cytokine secretion	[21]
	Th17 cells	cell line	10 μg/mL	histone demethylation	In vitro, VitC reduces H3K9me3 levels and upregulates IL17 expression in a JMJD2-dependent manner	[22]
	B cells	cell line and animal	0–20 μM (in vitro); 4 g/kg, i.p. (in vivo)	DNA demethylation	VitC promotes B-cell differentiation and humoral immunity via enhancing the enzymatic activity of TET2/3 in vitro and in vivo	[23]
	NK cells	cell line	50 ng/mL	DNA demethylation	Low-dose VitC promotes KIR promoter demethylation and KIR expression, representing the maturation of NK cells	[24]
	Monocytes	cell line	0–500 μM	-	VitC induces alterations in monocyte surface markers, gene expression and protein secretion in a mimicked hypoxic microenvironment (1% O_2_) in vitro	[25]
	Macrophages	cell line and animal	0–4 mM (in vitro); 2 g/kg, 4 g/kg, i.p. (in vivo)	-	High-dose VitC induces apoptosis of M2 macrophages in TME and dose-dependently inhibits EMT and metastasis in ovarian cancer	[26]
	Neutrophils	animal	0.33 g/L, oral; 200 mg/kg, i.p.	multi-pathways	Oral VitC attenuates NETs formation and autophagic gene expression as well as inhibits NF-κB activation	[27]
	Neutrophils	animal	4 g/kg, oral	-	Oral high-dose VitC prevents melanoma invasion and increases neutrophil infiltration within the tumor	[28]
	DCs	cell line and animal	0–2 mM (in vitro); 0.08 mM vcDC (in vivo)	signal molecules modulation	VitC increases IL-12 and IFN-γ secretion from DC cells, which in turn drives Th1 immunity	[29]
	Fibroblasts	cell line	0–20 μM	-	VitC regulates the expression of genes related to ECM remodeling and cell adhesion, thereby affecting the phenotype of immortalized MEF	[30]
	Endothelial cells	cell line	0–200 μM	multi-pathways	VitC improves endothelial cell dysfunction through multiple molecules, involving NO, ROS, RNS, biopterins, and GSH	[31]
	Neurons	cell line	200 μM	signal molecules modulation	VitC oxidation induces necrotic apoptosis of neurons in a ROS-independent manner	[32]

Abbreviations: CRC, colorectal cancer; VHL tumor suppressor protein; ccRCC, clear cell renal cell carcinoma; AML, acute myeloid leukemia; NK cells, natural killer cells; DCs, dendritic cells; EMT, epithelial–mesenchymal transition; NETs, neutrophil extracellular traps; ECM, extracellular matrix; 2-DG, 2-deoxy-glucose; i.p., intraperitoneal injection; VitC, vitamin C; pVC, phospho-modified vitamin C; IVC, intravenous vitamin C; vcDC, vitamin C-treated DCs vaccination.

**Table 2 cancers-14-02608-t002:** Preclinical and clinical evidence for different doses of VitC as a novel therapeutic adjuvant.

Combination Therapy		Study Type(s)	VitC Dose	Cancers (Cell Lines)	Target	Outcome(s)	Reference/NCT Number
Immunotherapy	ICT (anti-PD-1, anti-CTLA-4) + VitC i.p.	animal	4 g/kg	breast cancer (TS/A, 4T1), colorectal cancer (CT26, MC38), pancreatic cancer (PDAC), melanoma (B16-F10)	CD4^+^ and CD8^+^T lymphocytes, cancer cells	VitC increases the recruitment of lymphocytes in TME and improves the responsiveness of MMR-deficient tumors to ICT	[18]
	ICT (anti-PD-1) + VitC i.p.	cell line and animal	1 mM (in vitro); 4 g/kg (in vivo)	B-cell lymphoma (A20, SU-DHL-6, OCI-Ly1, OCI-Ly7, OCI-Ly3)	CD8^+^T cells, macrophages, cancer cells	VitC synergistically increases ICT efficacy by enhancing retrovirus expression, CTLs infiltration, and IL12 production in lymphoma	[35]
	ICT + IVC	cell line and animal	250 μM (in vitro); 4 g/kg (in vivo)	melanoma (B16-OVA), leukemia (THP-1), colorectal cancer (MC38)	CD3^+^T cells, CTLs, CD56^+^NK cells, cancer cells	VitC upregulates TET-mediated cytokine expression to activate the IFN-γ/JAK2/STAT1 pathway, enhancing TILs infiltration, as well as ICT efficacy	[34]
	ICT + VitC i.p.	cell line and animal	0.5 g/kg	renal cell carcinoma (Renca, 786-O, A498)	CD4^+^ and CD8^+^T lymphocytes, cancer cells	VitC improves ICT efficacy via upregulation of cytokine and chemokine levels in a TET2-dependent manner, and indirectly induces PD-L1 expression	[86]
	ICT + VitC	cell line	0–50 μM	pancreatic cancer (PANC-1, BxPC-3 and MIA PaCa-2)	cancer cells	VitC inhibits histone acetyltransferase 1, which in turn downregulates PD-1 mRNA expression	[87]
	ICD (oAds) + VitC i.p.	cell line and animal	2 mM (in vitro); 4 g/kg (in vivo)	colon cancer (CT26), breast cancer (4T1), hepatocellular carcinoma (Hepa1-6)	DC cells, CD8^+^T cells, CD4^+^ T cells, CD3^+^T cells	High-dose VitC and oAds exhibit a synergistic antitumor effect, with increased CD8^+^ T cells and DCs and decreased M2-type TAM cells in TME	[77]
	DC vaccines + VitC i.p.	cell line and animal	0–2 mM (in vitro); 0.08 mM (in vivo)	melanoma (B16F10)	DC cells, CD8^+^T cells, CD4^+^ T cells	VitC promotes the secretion of co-cultured CD4^+^, CD8^+^ T cells in vitro and induces protective antitumor immunity in mice	[88]
Small-molecule kinase inhibitors	PI3K inhibitor (buparlisib) + oral VitC	cell line and animal	0, 50, 100, 300 μM (in vitro); 3.3 g/L (animal)	TNBC (BT20, MDA-MB-453)	cancer cells	Synergistically, VitC enhanced KDM5-mediated histone H3K4 demethylation and boosted the efficacy of buparlisib	[89]
	sorafenib + IVC	cell line and clinical	0–20 mM (in vitro); 75 g/infusion (clinical)	hepatocellular carcinoma (Hep G2, SNU-449, HuH-7), breast cancer (T47D), pancreatic cancer (MIA PaCa2)	cancer cells, angiogenesis	IVC and low-dose sorafenib exhibit synergistic cytotoxicity to suppress cancer viability and metastasis	[90]
	erlotinib + gemcitabine + IVC	clinical, phase I	50, 75, 100 g/infusion	pancreatic cancer	cancer cells, angiogenesis	IVC is well tolerated with erlotinib and gemcitabine in patients with advanced cancer	[91]
	tyrosine kinase inhibitors (osimertinib or tarceva or iressa) + IVC	clinical, phase I/II	30 g/infusion	EGFR mutant NSCLC	cancer cells	-	NCT03799094
Monoclonal antibodies	bevacizumab+ Temozolomide + oral VitC	clinical, phase I	250 mg/d	recurrent high-grade glioma	cancer cells, angiogenesis	-	NCT01891747
	FOLFOXIRI +/- bevacizumab + IVC	clinical, phase III	1.5 g/kg	peritoneal metastatic colorectal cancer	cancer cells, angiogenesis	-	NCT04516681
	mFOLFOX6 +/- bevacizumab + IVC	clinical, phase III	1.5 g/kg	colorectal neoplasms	cancer cells, angiogenesis	-	NCT02969681
	cetuximab + VitC i.p.	cell line and animal	1 mM, 2 mM (in vitro); 4 g/kg (animal)	colon cancer (RAS/BRAF wt, DiFi, CCK81, C75, IRCC-10A)	cancer cells, angiogenesis	Combination therapy delays the emergence of acquired drug resistance in EGFR mutant tumors in vitro and in vivo	[92]
Metabolic inhibitors	antibiotics (doxycycline, azithromycin) + VitC	cell line	0–500 μM	breast cancer stem cells (MCF7)	cancer cell mitochondria	VitC and glycolysis inhibitor form a synthetic lethal strategy that targets both OXPHOS and glycolysis	[93,94]
	metformin + IVC	clinical, phrase II	1.5 g/kg	hepatocellular carcinoma, pancreatic cancer, gastric cancer, colorectal cancer	cancer cell mitochondria and other targets	-	NCT04033107
	glycolysis inhibitors (3-PO) + VitC	cell line	0–20 mM	NSCLC (H1299, H661, A549)	cancer cells	VitC synergizes with glycolysis inhibitors to induce apoptosis in NSCLC, mainly through the upregulation of ROS	[52]
Epigenetic therapies	DNMTis (5-aza-CdR) + VitC	cell line	57 μM	colorectal cancer (HCT116), APL (HL60), breast cancer (MCF7), liver cancer (HepG2, SNU398)	cancer cells	In cooperation with DNMTis, low-dose VitC acts as a TET enzyme stimulator, which enhances viral mimicry response via endogenous retroviral gene transcription	[95]
	DNMTis (5-azacytidine) + oral VitC	clinical	500 mg/d	AML, MDS, CMML	cancer cells	The treatment increased 5hmC/5mC levels in patients and upregulated retroviral gene expression in DNMTi naïve patients compared to the placebo group	NCT02877277; [76]
	BETi + oral VitC	cell line and animal	50–300 μM (in vitro); 3.3 g/L (in vivo)	TNBC (MDA-MB-231, BT-549, HCC1937), melanoma (A2058, SK-MEL28, SK-MEL2, C8161, 1205Lu)	cancer cells	Oral VitC and BETi collectively inhibit histone acetylation and improve tumor response to BETi treatment in vitro and in vivo. The underlying molecular mechanisms involve disruption of BRD4 and H4 interactions and upregulation of HDAC1 expression	[96,97]
Diet therapy	ketogenic diet + IVC	clinical	15–40 g/d	multiple cancers	cancer cells	VitC controls the inflammatory status of patients with advanced cancer, as well as increases ketone body content after a ketogenic diet	[98]
	fasting-mimicking + IVC	cell line and animal	350 μM (in vitro); 4 g/kg (in vivo)	KRAS mutant cancers: colorectal cancer (HCT116, DLD-1, CT26), lung cancer (H23, H727), pancreatic cancer (PANC1)	cancer cells	VitC and fasting-mimicking synergistically disrupt ROS and iron metabolism to enhance toxicity to KRAS-mutated tumor cells, sensitizing oxaliplatin therapy	[99]
	very low carbohydrate diet + IVC	clinical, phase I/II	25, 50, 75, 100 g/infusion	KRAS and BRAF mutant colon cancer stage IV	cancer cells	-	NCT04035096

Abbreviations: i.p., intraperitoneal injection; IVC, intravenous vitamin C; ICT, immune checkpoint therapy; PD-1, programmed death-1; CTLA-4, cytotoxic T-lymphocyte-associated antigen 4; ICD, immunogenic cell death; oAds, oncolytic adenoviruses; PI3K, phosphoinositide 3-kinase; FOLFOXIRI, (5-fluorouracil, leucovorin, oxaliplatin, and irinotecan); mFOLFOX6, (5-fluorouracil, leucovorin, oxaliplatin); 3-PO, 3-(3-pyridinyl)-1-(4-pyridinyl)-2-propen-1-one; DNMTis, DNA methyltransferase inhibitors; 5-aza-CdR, 5-aza-2′-deoxycytidine; BETi, bromodomain and extra-terminal domain inhibitors; NSCLC, non-small cell lung cancer; APL, acute promyelocytic leukemia; AML, acute myeloid leukemia; MDS, myelodysplastic syndromes; CMML, chronic myelomonocytic leukemia; TNBC, triple negative breast cancer; TILs, tumor-infiltrating lymphocytes; CTLs, cytotoxic T lymphocyte.

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
