# Peer review of "Repurposing Vitamin C for Cancer Treatment: Focus on Targeting the Tumor Microenvironment"

_cancers, 2022, doi:10.3390/cancers14112608_

Round 1
Reviewer 1 Report
1. Description of vitamin C transport mechanism into the cancer cells will be beneficial to the readers.
2. Many typographical and grammatical errors in this review article.
Author Response
Responses to Reviewer #1’s Comments
- “Description of vitamin C transport mechanism into the cancer cells will be beneficial to the readers.”
Authors’ response: Thank you for pointing out this point, and we add some related information and Ref [15] in the revised manuscript. Page 3 to 4
In terms of transport mechanisms, the transmembrane transporters SVCTs and GLUTs transport ascorbic acid and DHA, respectively, and promote intracellular accumulation in tumor cells and other normal cells. The expression levels of these two transporters vary by organ and correspond to plasma concentrations, metabolic needs, and some specific gene mutations [15].
- “Many typographical and grammatical errors in this review article.”
Authors’ response: We feel sorry for our carelessness. In our resubmitted manuscript, those typographical and grammatical errors are corrected. Also, we checked the revised manuscript with three native English speakers from a professional English editing company.
Reviewer 2 Report
In this manuscript, the authors have fully investigated the role of vitamin C in cancer treatment, especially its function in tumor microenvironment. They divided the whole complex TME into seven specialized microenvironments and discussed the possible role of vitamin C in each of them. Furthermore, they have discussed the application of vitamin C in cancer treatment as the single or adjuvant drug. Overall, they have provided the detailed information of vitamin C in tumor microenvironment.
Author Response
Responses to Reviewer #2’s Comments
“In this manuscript, the authors have fully investigated the role of vitamin C in cancer treatment, especially its function in tumor microenvironment. They divided the whole complex TME into seven specialized microenvironments and discussed the possible role of vitamin C in each of them. Furthermore, they have discussed the application of vitamin C in cancer treatment as the single or adjuvant drug. Overall, they have provided the detailed information of vitamin C in tumor microenvironment.”
Authors’ response: Thank you for the careful reading and commentary.
Reviewer 3 Report
This review summarizes the recent advances in deciphering the roles of Vitamin C in reshaping the tumor microenvironment (TME) and provides insights for the potential combination between Vitamin C and current novel therapies to treat multiple types of cancers. The paper is well organized, neatly phrased, and supported with beautiful and informative figures. It offers a complete and updated picture of how Vitamin C can be further exploited in current and future cancer therapies.
Some concerns or comments for the current version:
- The authors categorized the TME into seven types by their characteristics or components, instead of target organs. However, TME itself is more complicated in different types of organs, especially when it comes to the distal metastases. For example, bone microenvironment is considered as immunosuppressive, relatively hypoxic and permissive for certain types of primary tumors (e.g. prostate cancer) to colonize and metastasize. Vitamin C is vital for bone homeostasis. But the authors did not discuss a word about bone microenvironment. Considering the soaring needs in deciphering microenvironment for metastases, the authors are recommended to discuss the role of Vitamin C in some major type of organs (lung, bone, liver, etc.) that are prone to host metastases, thus to strengthen the potential of repurposing Vitamin C in cancer therapies.
- The authors mentioned the controversial debate over whether Vitamin C can combat cancer but did not present enough key points of summary for the opposing evidence. Since Vitamin C is newly emerged re-purposed cancer drug candidate, the authors are recommended to make some comments or insights about how this controversy might be resolved by which type of study. This will do no harm to the proposed title but show the critical thinking of the ideas instead.
- The authors used many abbreviations in the manuscript, and some of the items only appeared twice throughout and located far apart. Then spelling it out will relief the readers from the efforts to jump back and forth. For examples, RCT, FMD.
- Some contents need further elaboration to provide essential background information.
4.1 The target or mechanism for the mAb, Bevacizumab.
4.2 On page 9, “several studies have also focused on VitC enhancing the anti-tumor efficacy of epigenetic agents BETi and DNMTi [66, 82, 83]”. At least one of the studies referred here was performed in mice. Noting that TME in mouse and human might be different. It is important to address that these are pre-clinical studies in vitro or in vivo so that the readers will not over-interprete the claims in this manuscript. There are several other studies that also need a few more words to clarify.
4.3 Since this paper was submitted to Cancers and intended for a broad spectrum of readers, the authors should guide readers to grasp the key points from the evidence they cited. For example, a sentence on Page 7 “Otten et al. observed an increase in the relative abundance of Lachnospiraceae (p < 0.05), which also produces SCFA, and a decrease in the relative abundance of the Bacteroidetes (p < 0.01) and Enterococci (p < 0.01) [64]”. The authors did a good job in explaining the increase of Lachnospiraceae as a stimulus of SCFA production, then how about the decreases of Bacteroidetes and Enterococci?
4.4 Try not to use “affect” because this word does not specify whether it is a up-regulation or down-regulation.
In summary, this manuscript is well written and adds great value to the current cancer research field, and is recommended for publication on Cancers after the revisions mentioned above.
Author Response
Responses to Reviewer #3’s Comments
Firstly, we are grateful to the reviewer for his/her professional comments and valuable suggestions. Many of these suggestions contribute to the depth and implications of this paper.
- “The authors categorized the TME into seven types by their characteristics or components, instead of target organs. However, TME itself is more complicated in different types of organs, especially when it comes to the distal metastases. For example, bone microenvironment is considered as immunosuppressive, relatively hypoxic and permissive for certain types of primary tumors (e.g. prostate cancer) to colonize and metastasize. Vitamin C is vital for bone homeostasis. But the authors did not discuss a word about bone microenvironment. Considering the soaring needs in deciphering microenvironment for metastases, the authors are recommended to discuss the role of Vitamin C in some major type of organs (lung, bone, liver, etc.) that are prone to host metastases, thus to strengthen the potential of repurposing Vitamin C in cancer therapies.”
Authors’ response: We sincerely thank you for the constructive comments on this review. And we have written a new paragraph to better illustrate the role of VitC in this regard. Page 8 to 9
3.8. The interplay between VitC and the complicated metastatic TME
To be precise, the TMEs are more like a complex whole with distinct organ features rather than separated compartments. Metastatic tumor foci retain the characteristics of the primary tumor while also evolving to accommodate inefficient metastatic processes and new target organs. For instance, bone is one of the most common sites of metastasis for multiple advanced tumors, exhibiting complex and unique features in immunosuppression, hypoxia, and ECM remodeling. There has been much evidence to support the role of VitC in bone metabolism, thus it is worth exploring whether VitC affects bone metastasis.
During bone metastases, cellular compounds produced by the primary tumor enter the bloodstream and act at distant sites, stimulating angiogenesis and infiltration of Tregs, MDSCs, and other immune cells to generate a pre-metastatic niche. The colonized tumor cells then continuously interact with microenvironments to promote skeletal remodeling and immunosuppression, culminating in overt bone metastases [70]. Also, hypoxia and ECM remodeling are engaged in the process. By epigenetic mechanisms, VitC significantly improves bone metabolism and bone formation. VitC modulates stromal cell differentiation, promoting osteoblast development while suppressing osteoclasts, possibly alleviating tumor-induced aggressive osteolysis [71, 72]. The synthesis of type I collagen is enhanced by VitC, which not only improves the stability of the bone matrix but also contributes to osteoblasts differentiation [73]. Furthermore, Kolke et al. demonstrated that higher doses of VitC (100 μM) altered miRNA expression to up- or down-regulate signaling pathways associated with cell adhesion, differentiation, and cell stemness in BMSCs [74]. RANKL/NFκB in osteoclasts is an essential signaling pathway that accelerates bone metastasis. Hie et al. reported that VitC deficiency status upregulates RANK expression in vitro and in vivo, suggesting a negative regulation of RANK [75]. These findings indicate that VitC may inhibit tumor growth by acting on metastases as well as the main location.
The existing research on the effects of VitC on the bone microenvironment focuses mostly on preventing osteolysis and bone loss. Further research into this function for certain osteogenic metastases, such as prostate cancer, is warranted [76]. Anyway, VitC may help patients with advanced cancers avoid complications like osteoporosis and fractures by acting as a supportive treatment.
- “The authors mentioned the controversial debate over whether Vitamin C can combat cancer but did not present enough key points of summary for the opposing evidence. Since Vitamin C is newly emerged re-purposed cancer drug candidate, the authors are recommended to make some comments or insights about how this controversy might be resolved by which type of study. This will do no harm to the proposed title but show the critical thinking of the ideas instead.”
Authors’ response: We agree that opposing evidence would make the article more rigorous. At this point, we add some negative results about VitC intake and the risk of breast cancer from Nurses' Health Studies. We note also that many previous studies have reported similar results. We added this section on page 3.
However, two Nurses' Health Studies conducted over 32 and 22 years, respectively, showed no association between VitC intake and risk of breast cancer. This finding is consistent with most previous prospective studies [8].
- “The authors used many abbreviations in the manuscript, and some of the items only appeared twice throughout and located far apart. Then spelling it out will relief the readers from the efforts to jump back and forth. For examples, RCT, FMD.”
Authors’ response: Thank you for your reminder. We have carefully revised the previous draft. On pages 3 and 8, we change the “RCT” to “randomized controlled trials”. On page 11, we replace “FMD” with “fasting-mimicking diet”, and we also replace “RCC” with “renal cell carcinoma” to make the article clear.
- “The target or mechanism for the mAb, Bevacizumab.”
Authors’ response: The mechanism is described on page 10 of the text.
Bevacizumab inhibits tumor growth by blocking the binding of VEGF to the vascular endothelial receptor, which targets the tumor vascular system.
- On page 9, “several studies have also focused on VitC enhancing the anti-tumor efficacy of epigenetic agents BETi and DNMTi [66, 82, 83]”. At least one of the studies referred here was performed in mice. Noting that TME in mouse and human might be different. It is important to address that these are pre-clinical studies in vitro or in vivo so that the readers will not over-interprete the claims in this manuscript. There are several other studies that also need a few more words to clarify.”
Authors’ response: Thank you for your advice, and we have added information about the types of studies associated with these results on page 10.
In mice bearing triple-negative breast cancer and melanoma, VitC supplementation improves sensitivity to BETi therapy [91, 92]. In a randomized, double-blind trial in patients with bone marrow cancer, oral VitC promoted DNA demethylation in a synergistic manner with DNMTi therapy (5-azacytidine) [68].
- “Since this paper was submitted to Cancers and intended for a broad spectrum of readers, the authors should guide readers to grasp the key points from the evidence they cited. For example, a sentence on Page 7 “Otten et al. observed an increase in the relative abundance of Lachnospiraceae (p < 0.05), which also produces SCFA, and a decrease in the relative abundance of the Bacteroidetes (p < 0.01) and Enterococci (p < 0.01) [64]”. The authors did a good job in explaining the increase of Lachnospiraceae as a stimulus of SCFA production, then how about the decreases of Bacteroidetes and Enterococci?”
Authors’ response: We apologize for some of the confusing points. Current studies have not clarified the role of decreased Bacteroidetes and Enterococci in the tumor. As a result, we've changed the statement's focus to make things clearer for the readers.
Unlike the above, Otten et al. did not directly assay gut metabolites, but they found a significant increase (P < 0.05) in the relative abundance of Lachnospiraceae in the flora, a microbe that also produces SCFAs [66].
- “Try not to use “affect” because this word does not specify whether it is an up-regulation or down-regulation.”
Authors’ response: Thank you for your suggestion, we have made the changes as follows.
On page 8, “Given that bacterial metabolites, such as SCFAs, can enhance the immunogenicity of situ and a variety of distant cancers, the results open new possibilities for VitC to enhance anticancer immunity [67].” We change “affect” to “enhance the immunogenicity of”.
On page 6, “VitC modulates synaptic plasticity and facilitates the storage and release of other neurotransmitters, such as dopamine, acetylcholine, glutamate, neuropeptides [47-49].” We change “regulates” to “facilitates”.
On page 7, “VitC can promote neural development, promote neurite growth, and mediate the metabolic coupling between glial cells and neurons through the AA cycle [54, 55].” We change “affect” to “promote”.